# Comparison of Postoperative Nausea and Vomiting Incidence between Remimazolam and Sevoflurane in Tympanoplasty with Mastoidectomy: A Single-Center, Double-Blind, Randomized Controlled Trial

**DOI:** 10.3390/medicina59071197

**Published:** 2023-06-25

**Authors:** Seung Cheol Lee, Ji Wook Jung, So Ron Choi, Chan Jong Chung, Tae Young Lee, Sang Yoong Park

**Affiliations:** Department of Anesthesiology and Pain Medicine, Dong-A University Hospital, 26 Daesingongwon-ro, Seo-gu, Busan 49201, Republic of Korea; k57501@dau.ac.kr (S.C.L.); wnr2749@gmail.com (J.W.J.); choisr@dau.ac.kr (S.R.C.); cjchung@dau.ac.kr (C.J.C.); eggrobo1024@gmail.com (T.Y.L.)

**Keywords:** general anesthesia, mastoidectomy, postoperative nausea and vomiting, remimazolam, sevoflurane, tympanoplasty

## Abstract

*Background and Objectives*: Postoperative nausea and vomiting (PONV) is a common adverse effect of general anesthesia, especially in middle ear surgery. Remimazolam is a newer benzodiazepine recently approved for use in general anesthesia. This study aimed to compare the incidence rate of PONV after tympanoplasty with mastoidectomy between using remimazolam and sevoflurane. *Materials and Methods*: This study included 80 patients undergoing elective tympanoplasty with mastoidectomy. The patients were randomly assigned to either the remimazolam or sevoflurane group. The primary outcome was the incidence rate of PONV 12 h after surgery. The secondary outcomes were the incidence rate of PONV 12–24 and 24–48 h after surgery, severity of PONV, incidence rate of vomiting, administration of rescue antiemetics, hemodynamic stability, and recovery profiles. *Results*: The incidence rate of PONV 0–12 h after tympanoplasty with mastoidectomy was significantly lower in the remimazolam group compared with that in the sevoflurane group (28.9 vs. 57.9%; *p* = 0.011). However, the incidence rate of delayed PONV did not differ between the two groups. PONV severity in the early periods after the surgery was significantly lower in the remimazolam group than in the sevoflurane group. The incidence rate of adverse hemodynamic events was lower in the remimazolam group than in the sevoflurane group, but there was no difference in the overall trends of hemodynamic data between the two groups. There was no difference in recovery profiles between the two groups. *Conclusions*: Remimazolam can significantly reduce the incidence rate of early PONV after tympanoplasty with mastoidectomy under general anesthesia.

## 1. Introduction

Postoperative nausea and vomiting (PONV) is a common adverse effect that often occurs after surgery under general anesthesia. PONV occurs in approximately 30% of all postsurgical patients who receive general anesthesia, and its incidence rate can be up to 80% in high-risk patients (patients who are female, are nonsmokers, have a history of PONV, and receive postoperative opioids) [1,2,3,4]. Furthermore, owing to its surgical site characteristics, the incidence rate of PONV in patients who undergo tympanoplasty with mastoidectomy, which is a middle ear surgery, is 50–80% [5,6]. The perioperative management of PONV is important because its occurrence may be related to increased length of stay in the postanesthesia care unit (PACU), potential hospital admission, total healthcare cost, and patient dissatisfaction [7]. In this context, considering the high incidence rate of PONV after tympanoplasty with mastoidectomy, the prophylactic use of antiemetics to prevent PONV is common. Several antiemetics, including 5-hydroxytryptamine 3 receptor antagonists, dopamine receptor antagonists, corticosteroids, and antihistamines, have been used for the prevention and treatment of PONV. However, each of these drugs has limitations; thus, none of these antiemetics can entirely prevent and treat PONV after middle ear surgery [8].

Factors related to general anesthesia that can alter PONV incidence include intraoperative anesthetic agents. Volatile anesthetic use is an important factor that causes PONV after surgery [9,10]. Subsequently, several studies have carried out to confirm that using intravenous drugs for total intravenous anesthesia or combination with volatile anesthetics rather than volatile anesthetics alone reduces the incidence rate of PONV [9,10,11,12,13].

Remimazolam besylate is a newly developed ultra-short-acting benzodiazepine that can be used for the induction and maintenance of general anesthesia as an intravenous anesthetic. It has several advantages, such as the following: it has a quick onset and offset, results in hemodynamic stability, and existence of reversal agent, namely, flumazenil [14,15,16]. Remimazolam has a similar chemical structure to that of midazolam, and the ester-linked side chain attached to diazepine ring makes this drug more rapidly metabolized [17,18]. Its sedative effect comes from a specific gamma aminobutyric acid subtype A (GABA_A_) receptor, which is a similar receptor for midazolam [17,19].

The administration of intravenous midazolam significantly reduces the overall incidence rate of PONV after middle ear surgery [6,13]. However, there are no previous studies on the anesthetic technique using remimazolam for the prevention of PONV after tympanoplasty with mastoidectomy. Considering that remimazolam has similar structure to that of midazolam [17,20,21], as both are benzodiazepines, we hypothesized that intravenous anesthesia using remimazolam may lower the incidence rate of PONV after middle ear surgery under general anesthesia. This study aimed to compare the incidence rate of PONV between using remimazolam and sevoflurane as general anesthetic agents in patients who underwent tympanoplasty with mastoidectomy.

## 2. Materials and Methods

This prospective, double-blind, randomized trial was approved by the Institutional Review Board and was registered at the Clinical Research Information Service on cris.nih.go.kr. Before enrollment, written informed consent was obtained from all patients who agreed to participate in this study. This study was conducted in agreement with the principles described in the Declaration of Helsinki.

The patients enrolled in this study were people aged between 20 and 80 years old who were scheduled to undergo middle ear surgery under general anesthesia at our hospital from February to October 2022 and had American Society of Anesthesiologists physical status class I, II, or III. Patients hypersensitive to remimazolam and other benzodiazepines, previously treated with corticosteroids within 6 months, undergoing emergency surgery, with morbid obesity (body mass index (BMI) > 35 kg/m^2^), in the intensive care unit or previously scheduled to be transferred to the intensive care unit after surgery, who were pregnant, with cognitive impairment, and who did not provide consent were excluded from this study. Patients’ risk factors for PONV were assessed based on Apfel score (female, nonsmoking, with history of motion sickness or PONV, and postoperative use of opioids) using patient questionnaire and medical record [1].

The patients were randomly allocated to either the remimazolam (Byfavo^®^, Hana, Seoul, Republic of Korea) or sevoflurane (Sevofran^®^, Hana, Seoul, Republic of Korea) group in a 1:1 ratio in double-blind manner using computer-generated randomization software (nQuery Advisor^®^ version 7.0; Statsols, BMDP Statistical Software Inc., Cork, Ireland) and the sealed envelope method. An assistant, not involved in the anesthesia of the patients, unsealed the envelope that had group allocation and prepared remimazolam or sevoflurane for this study. To ensure blinding, one anesthesiologist induced the anesthesia, and another anesthesiologist, who did not participate in the induction of anesthesia to the patients, assessed the outcomes of this study. Another investigator evaluated the overall results of this study. The investigators, outcome assessors, statistician, and patients were fully blinded to group identity, whereas the attending anesthesiologists could not be blinded because of the different characteristics of the two anesthetics.

No premedication was administered prior to anesthesia. When entering the operating room, the patients were continuously monitored using electrocardiography, noninvasive arterial blood pressure, pulse oximetry, capnography, and bispectral index (BIS) monitoring. Nasopharyngeal temperature was also monitored and maintained at 36.5 ± 0.5 °C during surgery.

The patients assigned to the remimazolam group received continuous infusion of remimazolam at a rate of 12 mg/kg/h and remifentanil at 0.5 mcg/kg/min for induction. The infusions were continued until the confirmation of loss of consciousness (LoC), which was clarified by loss of verbal response and eyelash reflex. After LoC, endotracheal intubation was facilitated using rocuronium (0.8 mg/kg). The patients were mechanically ventilated with a tidal volume of 6–8 mL/kg using 40% oxygen, and air mixture and respiratory rate were adjusted to maintain an end-tidal CO_2_ (EtCO_2_) between 30 and 35 mmHg. The anesthesia was maintained using continuous infusion of remimazolam at a rate of 1–2 mg/kg/h and remifentanil at 0.1–1.0 mcg/kg/min according to hemodynamic responses and BIS value, which was maintained between 40 and 60 throughout the surgery.

In the sevoflurane group, anesthesia was induced with continuous infusion of remimazolam at a rate of 12 mg/kg/h and remifentanil at 0.5 mcg/kg/min for induction. After confirming LoC, the patients were intubated using an endotracheal tube after the administration of rocuronium (0.8 mg/kg). The patients were mechanically ventilated with a tidal volume of 6–8 mL/kg using 40% oxygen, and air mixture and respiratory rate were adjusted to maintain an EtCO_2_ between 30 and 35 mmHg. The anesthesia was maintained using sevoflurane at 1.5–2.5 vol% and continuous infusion of remifentanil at 0.1–1.0 mcg/kg/min. If necessary, an additional single intravenous bolus administration of remifentanil at a dosage of 0.5–1.0 mcg/kg was allowed in both groups. Rocuronium was appropriately administered for additionally needed neuromuscular blockade. At the end of the surgery, all anesthetics were simultaneously discontinued, and 3 mg/kg of sugammadex was intravenously administered at a train-of-four ratio of >90% to reverse residual neuromuscular blockade. Endotracheal extubation was performed under confirmation of recovery of self-respiration, defined as sufficient spontaneous breathing. The patients were transferred to the PACU, and the time until eye opening, cooperation, and Aldrete score reaching >9 were measured.

Hypotension was defined as decreased mean arterial pressure (MAP), <65 mmHg, which was calculated automatically during monitoring, and when it occurred, phenylephrine 100 mcg or ephedrine 10 mg was adequately administered with or without loading of fluid volume (100 mL). Bradycardia and tachycardia were defined as decreased heart rate (HR), <50 beats per minute (bpm), and increased HR, >100 bpm, respectively. For postoperative analgesia, ibuprofen 400 mg was intravenously administered routinely within approximately 6 h of surgery. Intravenous ketorolac 30 mg was administered when the numerical rating scale (NRS) of pain intensity exceeded 4 and when the patient required pain relief. Opioids, including fentanyl, were not postoperatively administered. Ramosetron (Nasea^®^, Astellas, Tokyo, Japan) 0.3 mg was intravenously administered as a rescue antiemetic if patients requested a treatment for PONV.

Every assessment was performed by the anesthesiologists, who were blinded to the allocated groups and did not participate in the induction of anesthesia to the patients. The primary outcome of this study was the incidence rate of PONV 0–12 h after middle ear surgery under general anesthesia. One or more episodes of nausea or vomiting were considered PONV. Nausea was defined as the subjectively unpleasant feeling associated with the perception of the urge to vomit, and vomiting was defined as the forceful expulsion of gastric contents from the mouth. The incidence rate of PONV was estimated according to the questionnaire provided to the patients.

The secondary outcomes were the incidence rate of PONV 12–24 and 24–48 h after surgery; severity of PONV, which was measured using the nausea score (0: no nausea; 1: mild nausea; 2: moderate nausea; 3: severe nausea) 0–12, 12–24, and 24–48 h after surgery; incidence of vomiting 0–48 h after surgery; administration of rescue antiemetics; postoperative pain score; need of rescue analgesics; incidence rate of adverse hemodynamic events, including hypotension and bradycardia and/or tachycardia; total amount of intravenous phenylephrine and ephedrine use; hemodynamic data, including MAP and HR before induction (T0), 1 min after induction (T1), 30 min after induction (T2), 1 min before emergence (T3), 1 min after emergence (T4), and 30 min after emergence (T5); Modified Observer’s Alertness/Sedation Scale (MOAA/S) score at T0, T1, T2, T3, T4, and T5; total amount of anesthetics and remifentanil administered throughout surgery; time to extubation; time to eye opening; time to cooperation; time to Aldrete score > 9; duration of hospital stay time after surgery; and postoperative complications, including headache, dizziness, respiratory complications such as hypoxia, pulmonary edema, and bronchospasm, sleep disorder, amnesia, and delirium. All recorded parameters were evaluated within 48 h of surgery.

Sample size was calculated using nQuery Advisor^®^ version 7.0 (Statsols, BMDP Statistical Software Inc., Cork, Ireland). According to previous studies, the incidence rate of PONV after general anesthesia using sevoflurane is approximately 60% [5,6,11], and the administration of intravenous midazolam, which is a benzodiazepine similar to remimazolam, results in a lower incidence rate of PONV by up to 40% compared with inhalational anesthesia using sevoflurane alone [5,6]. Therefore, we anticipated that a 40% lower incidence rate of PONV after tympanoplasty with mastoidectomy when using intravenous remimazolam compared with when using volatile sevoflurane would be clinically significant. The α and β errors were set at 0.05 (two-sided) and 0.1 (power = 0.9), respectively. Upon this analysis, 34 patients were required for each group. Assuming a dropout rate of 15%, 40 patients per group would be required.

Variables are summarized by frequency and percentage for categorical data and mean ± standard deviation. Group differences were assessed using the chi-squared or Fisher’s exact test for categorical data and independent *t*-test or Mann–Whitney U test for numerical data, as appropriate. To assess if the distribution was normal, we used the Shapiro–Wilk test. Absolute standardized differences comparing the baseline covariates between groups were also reported. Considering the nature of the repeated measured data, a generalized linear mixed model (GLMM) with random intercepts was used to fit a model. The GLMM model included repeated measures of numerical variables as dependent variables; group, time, and group × time interaction as fixed effects; and subject as a random effect. To avoid making any assumptions about the covariance structure, we used an unstructured covariance matrix that was allowed to differ across groups for GLMM analysis. All statistical analyses were performed using SPSS version 26.0 statistical software (IBM Corp. Released 2019. IBM SPSS Statistics for Windows, version 26.0. Armonk, NY, USA: IBM Corp) and R statistical software (version 3.4.0; R Foundation, Vienna, Austria, http://www.r-project.org/ accessed on 1 November 2022). *p* values < 0.05 were considered statistically significant.

## 3. Results

During the study period, 80 patients were assessed for eligibility. Among these patients, two did not meet the inclusion criteria owing to morbid obesity (BMI > 35 kg/m^2^) and two did not provide consent to participate in this study. Overall, the remaining 76 patients were randomly and equally allocated to each group. All of these patients completed this study and were analyzed (Figure 1).

The demographic characteristics and intraoperative data of the patients are presented in Table 1. The patient characteristics were well balanced between the two groups. There were no statistically significant differences in parameters that determined PONV risk score between the two groups. The total amount of remifentanil administered throughout the surgery was significantly higher in the remimazolam group than in the sevoflurane group (0.6 ± 0.4 vs. 0.2 ± 0.2 mg, *p* < 0.001). The incidence rate of hypotensive events that occurred during intraoperative periods was lower in the remimazolam group than in the sevoflurane group (*p* < 0.001). The total amounts of phenylephrine and ephedrine administered throughout the surgery were lower in the remimazolam group than in the sevoflurane group (0.0 ± 0.0 vs. 50.0 ± 98.0 mcg, *p* = 0.002 and 0.3 ± 1.6 vs. 3.9 ± 6.4 mg, *p* < 0.001, respectively).

The incidence rate of PONV at each time point after tympanoplasty with mastoidectomy is depicted in Table 2. The incidence rate of PONV 0–12 h after the middle ear surgery, which was the primary outcome of this study, was significantly lower in the remimazolam group compared with that in the sevoflurane group (28.9 vs. 57.9%, *p* = 0.011). However, no statistically significant differences were observed in the incidence rate of PONV 12–24 and 24–48 h after the surgery between the two groups. Similarly, the severity of nausea 0–12 h after middle ear surgery was significantly lower in the remimazolam group compared with the sevoflurane group (*p* = 0.033), whereas there were no differences between the two groups 12–24 and 24–48 h after the surgery. The incidence rate of vomiting showed no differences between the two groups. Six patients in the sevoflurane group requested antiemetic rescue for PONV treatment, whereas in the remimazolam group, only two patients required antiemetics. However, this difference was not statistically significant.

The perioperative hemodynamic data and sedation scale scores are presented in Figure 2 and Table 3. Overall trends in MAP, HR, and MOAA/S score during the perioperative period were not significantly different between the two groups.

Profiles of patients’ recovery and postoperative outcomes are shown in Table 4. No statistically significant differences were observed in the time required for extubation, eye opening, cooperation, and Aldrete score > 9 after the cessation of the anesthetics between the two groups. The length of hospital stay time also showed no statistical difference between the two groups. Postoperative complications, including headache, dizziness, respiratory complications, and sleep disorder, were not significantly different between the two groups. None of the patients in both groups experienced amnesia and delirium. Postoperative pain score and the number of patients who requested rescue analgesics were also not statistically significantly different between the two groups. No other adverse events, including administration of flumazenil, were recorded during the study period in either group.

## 4. Discussion

This study aimed to compare the incidence rate of PONV 0–12 h after tympanoplasty with mastoidectomy under general anesthesia between using intravenous remimazolam and volatile sevoflurane. Based on the study data, the incidence rate of PONV 12 h after the middle ear surgery was significantly lower in the remimazolam group compared with the sevoflurane group. Furthermore, PONV severity 0–12 h after the surgery was statistically lower in the remimazolam group than in the sevoflurane group. However, no statistically significant differences were found in the incidence rates and severity of PONV 12–24 and 24–48 h after surgery. Hence, the outcome of this study suggests that the administration of remimazolam rather than sevoflurane as an anesthetic reduces relatively early PONV, but not delayed PONV.

Factors that cause PONV are multifactorial [4]. The risk factors for PONV include female sex, nonsmoking status, previous history of motion sickness and/or PONV, and postoperative opioid use. Patients who have multiple risk factors are more likely to experience PONV [1,22]. Another risk factor for the increased incidence rate of PONV is surgical characteristic. In particular, middle ear surgery increases the incidence rate of PONV possibly due to increased pressure in the middle ear [23]. The anesthetic technique using volatile agent is also a risk factor for PONV [4,10]. Volatile anesthetics are emetogenic, and there are no significant differences among volatile agents, including sevoflurane, desflurane, isoflurane, enflurane, and halothane [9,10]. A study conducted by Lee et al. [11] demonstrated that the administration of propofol as a total intravenous anesthesia (TIVA) results in a lower incidence rate of PONV compared with using sevoflurane. Furthermore, the administration of midazolam, which is a benzodiazepine similar with remimazolam, decreases the incidence rate of PONV induced by its specific GABA_A_ receptor [13,17,19]. Remimazolam has similar properties to midazolam [17,20,21], which contributes to the expectation that remimazolam would decrease the incidence rate of PONV as midazolam does. However, according to a study conducted by Apfel et al. [9], volatile anesthetics contribute to early PONV, but not delayed PONV. Moreover, a study conducted by Tramer et al. [12] confirmed that TIVA with propofol, rather than volatile anesthetics, may have clinical relevance with decreased incidence rate of PONV only in short term. Consistent with these previous studies, our study outcomes demonstrate that the administration of remimazolam results in a lower incidence rate of early PONV compared with the administration of sevoflurane. This result is thought to be due to the pharmacological properties of remimazolam. Remimazolam has pharmacodynamic characteristics typical of other benzodiazepines, but it has highly organ-independent elimination clearance. Compared with midazolam, remimazolam does not induce prolonged sedative effects after drug cessation, as it is rapidly metabolized by nonspecific tissue esterases to CNS7054, an inactive metabolite with a 300 times decreased binding affinity at the GABA_A_ receptor and with no clinically significant effect at the receptor site [19,21]. Accordingly, because remimazolam is an ultra-short-acting benzodiazepine that is rapidly metabolized, its effects on reducing the incidence rate of PONV might be limited to only early postoperative periods. Further studies with more detailed and segmented time points would be required to accurately define early PONV and the antiemetic effect of remimazolam.

The incidence of PONV is associated with postoperative pain and perioperative use of opioids [22]. In this study, there were no significant differences in postoperative NRS pain score and the need of rescue analgesics between the two groups. This outcome might have been affected by the differences in the total dose of remifentanil administered during surgery. The total amount of remifentanil administered throughout the surgery was significantly higher in the remimazolam group compared with that in the sevoflurane group. This could be due to the differences in analgesic properties between benzodiazepine and sevoflurane. Sevoflurane has an analgesic effect [24], whereas the analgesic effect of benzodiazepine is unclear [25]. In contrast, Xie et al. [26] demonstrated that remimazolam has an analgesic potency and attenuates pain intensity. Therefore, more advanced studies are required to determine the analgesic effects of remimazolam. The increment in total amount of remifentanil administered during surgery contributes to the increased incidence rate of PONV after surgery. Hence, differences in remifentanil use between the remimazolam and sevoflurane groups might have affected the incidence rate of PONV in each group. Despite the fact that a larger amount of remifentanil use in the remimazolam group would increase PONV risk, the incidence rate of early PONV was lower in the remimazolam group than in the sevoflurane group in this study. This outcome may support the notable effect of remimazolam in decreasing the incidence rate of early PONV after surgery.

There are some studies that demonstrate the efficacy and safety of remimazolam in terms of hemodynamic stability [14,15,19]. Dhande et al. [27] reported that sevoflurane seemed to have greater hemodynamic stability compared with intravenous propofol, therefore demonstrating that it may be an appropriate anesthetic agent. Considering our study findings, the incidence rate of hypotensive events during the perioperative periods was significantly lower in the remimazolam group compared with that in the sevoflurane group. The total amounts of phenylephrine and ephedrine that were administered in terms of rescue were also lower in the remimazolam group than in the sevoflurane group. Based on these outcomes, we believe that remimazolam is superior to sevoflurane in terms of hemodynamic stability. However, the remimazolam group showed no differences in overall hemodynamic data trends compared to the sevoflurane group. It may have been affected by the immediate administration of phenylephrine or ephedrine under close monitoring following the hypotensive events. Although the incidence rate of hypotensive events was high in the sevoflurane group, the total amounts of administered phenylephrine and ephedrine were also high in the sevoflurane group. Therefore, the overall trends of hemodynamic data might have shown no differences between the two groups. Consequently, remimazolam can be considered a suitable anesthetic agent in terms of hemodynamic stability.

Although the incidence rate of early PONV was lower in the remimazolam group compared to the sevoflurane group, the incidence rate of 28.9% could be considered too high to justify the routine use of remimazolam to reduce the incidence of PONV. In future studies, it is considered necessary to compare the incidence of PONV between remimazolam and other anesthetics, including intravenous anesthetics. To the best of our knowledge, few studies have compared the incidence of PONV between remimazolam and other intravenous anesthetics. According to Kim et al. [28], there was no significant difference in the incidence and the severity of PONV between using remimazolam and propofol as intravenous anesthetics. Further studies will need to be conducted to justify the usefulness of remimazolam in reducing the incidence of PONV compared to other anesthetics.

Our study has some limitations. First, the sample size of this study was relatively small. Thus, further studies with larger sample sizes are required to better confirm the effects of remimazolam in decreasing the incidence rate of early PONV after surgery. Second, we were not able to confirm whether the depth of anesthesia in the remimazolam group was appropriate during the surgery. The BIS is a safe and effective method for monitoring anesthetic depth [29]. However, the accuracy of BIS as an indicator of anesthetic depth when administering remimazolam as an anesthetic agent remains controversial [15,30]. Shirozu et al. [30] reported that the remimazolam group had relatively higher BIS scores throughout the surgery compared with other anesthetics. Thus, additional studies should be conducted to determine the optimal method for evaluating anesthetic depth. Since we were not convinced whether the evaluation of anesthetic depth using BIS was appropriate, it may have affected the hemodynamic data and the use of vasoactive agent, which are related to the depth of anesthesia. Nevertheless, the protocol of this study is anticipated to be effective and safe, in that recovery time and MOAA/S after emergence from general anesthesia were rational. Finally, due to the different characteristics of the two anesthetics, attending anesthesiologist involved in anesthesia could not be blinded. Since the performing anesthesiologist was not blinded, it may have affected the amount of each anesthetic used, which in turn may have affected the depth of anesthesia and hemodynamic stability.

## 5. Conclusions

Despite being a novel drug, remimazolam is now widely used clinically owing to its utility and advantages, including the following: it has a rapid onset and offset, results in hemodynamic stability, and has a specific antagonist [14,15,16]. Apart from these advantages, our study suggests that remimazolam reduces the incidence rate of early PONV after surgery. In conclusion, intravenous remimazolam rather than volatile sevoflurane is a useful anesthetic agent induced to reduce the incidence rate of early PONV after tympanoplasty with mastoidectomy.

## Figures and Tables

**Figure 1 medicina-59-01197-f001:**
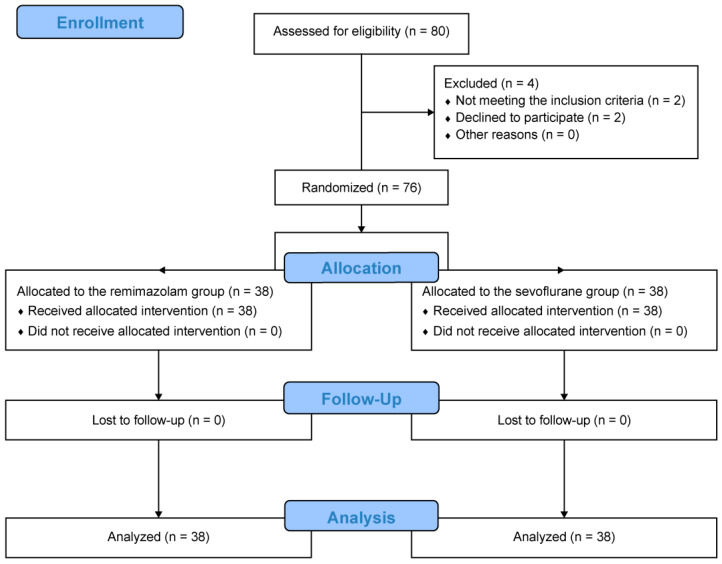
Consolidated Standards of Reporting Trials diagram depicting the study protocol of the randomized trial.

**Figure 2 medicina-59-01197-f002:**
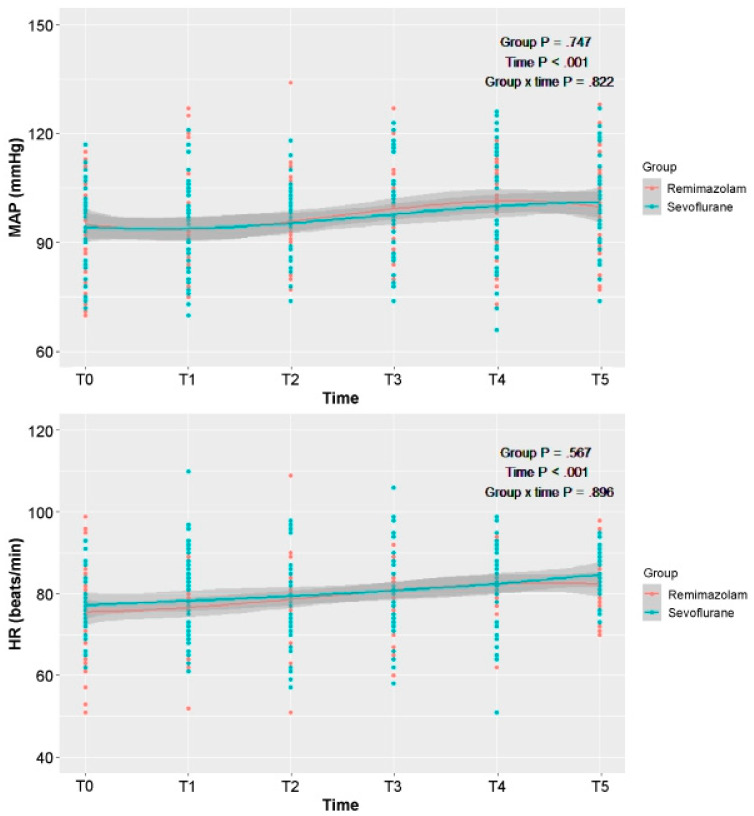
Perioperative hemodynamic variables, including MAPand HR. T0: before induction; T1: immediately after induction; T2: 30 min after induction; T3: before emergence; T4: immediately after emergence; T5: 30 min after emergence. MAP: mean arterial pressure; HR: heart rate.

**Table 1 medicina-59-01197-t001:** Demographic characteristics and intraoperative data.

Variable		Group		Absolute Standardized Difference	*p* Value
	Overall (n = 76)	Remimazolam (n = 38)	Sevoflurane (n = 38)		
Sex					
Male	33 (43.4)	18 (47.4)	15 (39.5)	0.16	0.488 ^3^
Female	43 (56.6)	20 (52.6)	23 (60.5)		
Age (yr)					
Mean ± SD	54.8 ± 13.2	55.7 ± 11.4	53.9 ± 14.9	0.13	0.856 ^2^
Median (IQR)	59.0 (50.0–63.0)	59.5 (50.0–63.0)	58.0 (50.0–64.0)		
Height (cm)					
Mean ± SD	162.1 ± 9.6	161.9 ± 8.4	162.2 ± 10.7	0.04	0.867 ^1^
Median (IQR)	162.4 (153.5–169.7)	162.6 (154.2–167.3)	162.3 (153.0–170.8)		
Weight (kg)					
Mean ± SD	65.9 ± 12.6	65.0 ± 10.5	66.8 ± 14.4	0.15	0.938 ^2^
Median (IQR)	63.5 (57.1–71.0)	63.9 (57.9–70.9)	62.6 (56.8–71.9)		
BMI (kg/m^2^)					
Mean ± SD	24.9 ± 3.0	24.7 ± 3.0	25.2 ± 3.0	0.14	0.541 ^1^
Median (IQR)	24.8 (23.4–26.5)	24.7 (23.4–26.4)	25.0 (23.3–26.6)		
ASA physical status					
1	21 (27.6)	12 (31.6)	9 (23.7)	0.18	0.792 ^4^
2	47 (61.8)	22 (57.9)	25 (65.8)	0.16	
3	8 (10.5)	4 (10.5)	4 (10.5)	0.00	
Risk factors for PONV					
Smoking history					
Yes	12 (15.8)	7 (18.4)	5 (13.2)	0.14	0.529 ^3^
No	64 (84.2)	31 (81.6)	33 (86.8)		
History of motion sickness and/or PONV					
Yes	0 (0.0)	0 (0.0)	0 (0.0)	0.00	-
No	76 (100.0)	38 (100.0)	38 (100.0)		
Postoperative opioid use					
Yes	0 (0.0)	0 (0.0)	0 (0.0)	0.00	-
No	76 (100.0)	38 (100.0)	38 (100.0)		
Total score (Apfel score)					
0	9 (11.8)	5 (13.2)	4 (10.5)	0.08	0.719 ^4^
1	27 (35.5)	15 (39.5)	12 (31.6)	0.16	
2	40 (52.6)	18 (47.4)	22 (57.9)	0.21	
3	(0.0)	(0.0)	(0.0)	0.00	
4	(0.0)	(0.0)	(0.0)	0.00	
Comorbidity					
HTN					
Yes	19 (25.0)	10 (26.3)	9 (23.7)	0.06	0.791 ^3^
No	57 (75.0)	28 (73.7)	29 (76.3)		
DM					
Yes	15 (19.7)	7 (18.4)	8 (21.1)	0.07	0.773 ^3^
No	61 (80.3)	31 (81.6)	30 (78.9)		
Diagnosis					
Cholesteatoma					
Yes	22 (28.9)	12 (31.6)	10 (26.3)	0.12	0.613 ^3^
No	54 (71.1)	26 (68.4)	28 (73.7)		
COM					
Yes	54 (71.1)	26 (68.4)	28 (73.7)	0.12	0.613 ^3^
No	22 (28.9)	12 (31.6)	10 (26.3)		
Duration of anesthesia (min)					
Mean ± SD	174.5 ± 37.5	172.0 ± 35.7	177.0 ± 39.6	0.13	0.539 ^2^
Median (IQR)	172.5 (150.0–195.0)	170.0 (150.0–195.0)	180.0 (149.0–200.0)		
Duration of surgery (min)					
Mean ± SD	138.1 ± 34.5	138.7 ± 31.8	137.6 ± 37.4	0.03	0.795 ^2^
Median (IQR)	140.0 (115.0–155.0)	140.0 (119.0–155.0)	140.5 (109.0–156.0)		
Fluid volume (mL)					
Mean ± SD	975.7 ± 143.6	973.7 ± 144.1	977.6 ± 145.1	0.03	0.906 ^1^
Median (IQR)	950.0 (900.0–1050.0)	950.0 (900.0–1060.0)	950.0 (887.5–1060.0)		
Total amount of remifentanil (mg)					
Mean ± SD	0.4 ± 0.4	0.6 ± 0.4	0.2 ± 0.2	1.13	<0.001 ^2^
Median (IQR)	0.3 (0.2–0.5)	0.5 (0.3–0.8)	0.2 (0.1–0.2)		
Total amount of remimazolam (mg)					
Mean ± SD	110.4 ± 111.4	209.7 ± 69.9	11.1 ± 2.8	4.02	<0.001 ^2^
Median (IQR)	69.5 (11.0–198.0)	198.0 (159.0–240.0)	11.0 (9.0–12.0)		
Adverse hemodynamic events					
Number of hypotension					
0	59 (77.6)	37 (97.4)	2 2(57.9)	1.06	<0.001 ^4^
1	5 (6.6)	1 (2.6)	4 (10.5)	0.32	
2	2 (2.6)	0 (0.0)	2 (5.3)	0.33	
3	5 (6.6)	0 (0.0)	5 (13.2)	0.54	
4	4 (5.3)	0 (0.0)	4 (10.5)	0.48	
5	1 (1.3)	0 (0.0)	1 (2.6)	0.23	
Number of bradycardia or tachycardia					
0	73 (96.1)	37 (97.4)	36 (94.7)	0.13	1.000 ^4^
1	3 (3.9)	1 (2.6)	2 (5.3)		
Total amount of phenylephrine (mcg)					
Mean ± SD	25.0 ± 73.3	0.0 ± 0.0	50.0 ± 98.0	0.72	0.002 ^2^
Median (IQR)	0.0 (0.0–0.0)	0.0 (0.0–0.0)	0.0 (0.0–25.0)		
Total amount of ephedrine (mg)					
Mean ± SD	2.1 ± 5.0	0.3 ± 1.6	3.9 ± 6.4	0.79	<0.001 ^2^
Median (IQR)	0.0 (0.0–0.0)	0.0 (0.0–0.0)	0.0 (0.0–10.0)		

Data are presented as mean ± SD, median (interquartile range), or number of patients (%), unless otherwise indicated. ^1^
*p* values were derived from the independent *t*-test. ^2^
*p* values were derived from Mann–Whitney’s U test. ^3^
*p* values were derived from the chi-squared test. ^4^
*p* values were derived from Fisher’s exact test. The Shapiro–Wilk test was used to test the normality assumption. BMI: body mass index; ASA: American Society of Anesthesiologists; PONV: postoperative nausea and vomiting; HTN: hypertension; DM: diabetes mellitus; COM: chronic otitis media; SD: standard deviation.

**Table 2 medicina-59-01197-t002:** Postoperative values.

Variable		Group		*p* Value
	Overall (n = 76)	Remimazolam (n = 38)	Sevoflurane (n = 38)	
0–12 h				
PONV				
Yes	33 (43.4)	11 (28.9)	22 (57.9)	0.011 ^1^
No	43 (56.6)	27 (71.1)	16 (42.1)	
PONV severity				
0	43 (56.6)	27 (71.1)	16 (42.1)	0.033 ^2^
1	12 (15.8)	6 (15.8)	6 (15.8)	
2	13 (17.1)	3 (7.9)	10 (26.3)	
3	8 (10.5)	2 (5.3)	6 (15.8)	
12–24 h				
PONV				
Yes	10 (13.2)	4 (10.5)	6 (15.8)	0.497 ^1^
No	66 (86.8)	34 (89.5)	32 (84.2)	
PONV severity				
0	66 (86.8)	34 (89.5)	32 (84.2)	0.791 ^2^
1	5 (6.6)	2 (5.3)	3 (7.9)	
2	5 (6.6)	2 (5.3)	3 (7.9)	
3	0 (0.0)	0 (0.0)	0 (0.0)	
24–48 h				
PONV				
Yes	2 (2.6)	1 (2.6)	1 (2.6)	1.000 ^2^
No	74 (97.4)	37 (97.4)	37 (97.4)	
PONV severity				
0	74 (97.4)	37 (97.4)	37 (97.4)	1.000 ^2^
1	2 (2.6)	1 (2.6)	1 (2.6)	
2	0 (0.0)	0 (0.0)	0 (0.0)	
3	0 (0.0)	0 (0.0)	0 (0.0)	
0–48 h				
Vomiting				
Yes	3 (3.9)	1 (2.6)	2 (5.3)	1.000 ^2^
No	73 (96.1)	37 (97.4)	36 (94.7)	
Antiemetic rescue				
Yes	8 (10.5)	2 (5.3)	6 (15.8)	0.262 ^2^
No	68 (89.5)	36 (94.7)	32 (84.2)	

Data are presented as mean ± SD, median (interquartile range), or number of patients (%), unless otherwise indicated. ^1^
*p* values were derived from the chi-square test. ^2^
*p* values were derived from Fisher’s exact test. The Shapiro–Wilk test was used to test the normality assumption. PONV: postoperative nausea and vomiting; SD: standard deviation.

**Table 3 medicina-59-01197-t003:** Perioperative hemodynamic variables and sedation scale, including MAP, HR, and MOAA/S score.

Variable		Group			Analysis for Repeated Measures	
	Overall (n = 76)	Remimazolam (n = 38)	Sevoflurane (n = 38)	*p*	Source	*p* *
MAP (mmHg)						
T0	94.5 ± 11.8	94.9 ± 12.0	94.1 ± 11.8	0.765 ^1^	Group	0.747
T1	94.5 ± 15.0	94.4 ± 16.3	94.6 ± 13.7	0.674 ^2^	Time	<0.001
T2	94.4 ± 11.5	94.6 ± 11.8	94.2 ± 11.2	0.747 ^2^	Group × time	0.822
T3	99.2 ± 12.9	99.9 ± 13.4	98.4 ± 12.5	0.628 ^1^		
T4	100.6 ± 15.6	101.3 ± 14.4	99.8 ± 16.8	0.683 ^1^		
T5	100.4 ± 13.0	99.7 ± 13.0	101.0 ± 13.1	0.661 ^1^		
HR (beats/min)						
T0	76.2 ± 10.4	75.6 ± 12.4	76.9 ± 7.9	0.599 ^1^	Group	0.567
T1	77.8 ± 10.4	76.2 ± 9.2	79.4 ± 11.3	0.183 ^1^	Time	<0.001
T2	78.6 ± 11.9	79.3 ± 11.5	77.9 ± 12.3	0.625 ^1^	Group × time	0.896
T3	80.8 ± 10.1	80.3 ± 8.2	81.3 ± 11.8	0.868 ^2^		
T4	82.6 ± 9.6	82.6 ± 8.0	82.6 ± 11.1	0.972 ^1^		
T5	83.5 ± 6.7	82.3 ± 7.4	84.6 ± 5.6	0.135 ^1^		
MOAA/S score						
T0	5.0 ± 0.0	5.0 ± 0.0	5.0 ± 0.0	1.000 ^2^	Group	0.965
T1	0.0 ± 0.0	0.0 ± 0.0	0.0 ± 0.0	1.000 ^2^	Time	<0.001
T2	0.0 ± 0.0	0.0 ± 0.0	0.0 ± 0.0	1.000 ^2^	Group × time	0.961
T3	0.0 ± 0.0	0.0 ± 0.0	0.0 ± 0.0	1.000 ^2^		
T4	3.2 ± 0.5	3.3 ± 0.6	3.2 ± 0.5	0.321 ^2^		
T5	4.9 ± 0.3	4.8 ± 0.4	4.9 ± 0.2	0.137 ^2^		

Data are presented as mean ± SD, unless otherwise indicated. ^1^
*p* values were derived from the independent *t*-test. ^2^
*p* values were derived from Mann–Whitney’s U test. * *p* values were derived from a generalized linear mixed model. The Shapiro–Wilk test was used to test the normality assumption. T0: before induction; T1: immediately after induction; T2: 30 min after induction; T3: before emergence; T4: immediately after emergence; T5: 30 min after emergence. MAP: mean arterial pressure; HR: heart rate; MOAA/S: Modified Observer’s Alertness/Sedation Scale; SD: standard deviation.

**Table 4 medicina-59-01197-t004:** Recovery profiles and postoperative outcomes.

Variable		Group		Absolute Standardized Difference	*p* Value
	Overall (n = 76)	Remimazolam (n = 38)	Sevoflurane (n = 38)		
Postoperative complications					
Headache					
Yes	32 (42.1)	13 (34.2)	19 (50.0)	0.32	0.163 ^3^
No	44 (57.9)	25 (65.8)	19 (50.0)		
Dizziness					
Yes	32 (42.1)	13 (34.2)	19 (50.0)	0.32	0.163 ^3^
No	44 (57.9)	25 (65.8)	19 (50.0)		
Respiratory complications					
Yes	3 (3.9)	1 (2.6)	2 (5.3)	0.13	1.000 ^4^
No	73 (96.1)	37 (97.4)	36 (94.7)		
Sleep disorder					
Yes	10 (13.2)	5 (13.2)	5 (13.2)	0.00	1.000 ^3^
No	66 (86.8)	33 (86.8)	33 (86.8)		
Amnesia					
Yes	0 (0.0)	0 (0.0)	0 (0.0)	0.00	-
No	76 (100.0)	38 (100.0)	3 8(100.0)		
Delirium					
Yes	0 (0.0)	0 (0.0)	0 (0.0)	0.00	-
No	76 (100.0)	38 (100.0)	38 (100.0)		
Postoperative pain (NRS)					
Mean ± SD	3.3 ± 2.0	3.1 ± 2.0	3.5 ± 2.1	0.22	0.409 ^2^
Median (IQR)	3.0(2.0–5.0)	3.0(2.0–5.0)	3.0(2.0–5.0)		
Need of rescue analgesics					
Yes	12	7 (18.4)	5 (13.2)	0.14	0.529 ^3^
No	64	31 (81.6)	33 (86.8)		
Time to extubation (min)					
Mean ± SD	8.3 ± 3.2	8.3 ± 3.4	8.3 ± 3.0	0.00	0.992 ^2^
Median (IQR)	8.0(6.0–10.0)	8.0(6.0–10.0)	8.0(6.0–10.0)		
Time to eye opening (min)					
Mean ± SD	10.6 ± 5.8	10.2 ± 5.3	11.1 ± 6.3	0.16	0.361 ^2^
Median (IQR)	9.0(7.0–12.0)	9.0(7.0–12.0)	9.5(8.0–12.0)		
Time to cooperation (min)					
Mean ± SD	21.7 ± 9.3	21.7 ± 6.3	21.7 ± 11.6	0.01	0.199 ^2^
Median (IQR)	20.0(16.0–24.0)	22.0(18.0–26.0)	19.0(15.0–23.0)		
Time to Aldrete ≥ 9 (min)					
Mean ± SD	30.5 ± 8.8	29.7 ± 7.9	31.2 ± 9.8	0.17	0.471 ^1^
Median (IQR)	29.5(26.0–35.0)	29.5(23.0–34.0)	29.5(27.0–36.0)		
Duration of postoperative hospital stay time (day)					
Mean ± SD	2.6 ± 1.4	2.6 ± 1.2	2.7 ± 1.6	0.06	0.386 ^2^
Median (IQR)	2.0(2.0–3.0)	2.0(2.0–3.0)	2.0(2.0–3.0)		

Data are presented as mean ± SD, median (interquartile range), or number of patients (%), unless otherwise indicated. ^1^
*p* values were derived from the independent *t*-test. ^2^
*p* values were derived from Mann–Whitney’s U test. ^3^
*p* values were derived from the chi-square test. ^4^
*p* values were derived from Fisher’s exact test. The Shapiro–Wilk test was used to test the normality assumption. NRS: numerical rating scale; SD: standard deviation.

## Data Availability

The data presented in this study are available on request from the corresponding author.

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
