# Peer review of "Comparison of Postoperative Nausea and Vomiting Incidence between Remimazolam and Sevoflurane in Tympanoplasty with Mastoidectomy: A Single-Center, Double-Blind, Randomized Controlled Trial"

_medicina, 2023, doi:10.3390/medicina59071197_

Round 1

Reviewer 1 Report

Reviewer report:

Comparison of postoperative nausea and vomiting incidence between remimazolam and sevoflurane in tympanoplasty with mastoidectomy: a single-center, double-blind, randomized con-trolled trial

Version 1, Date 31 May

Objective: 

The authors compared the incidence rate of PONV after tympanoplasty with mastoidectomy between using remimazolam and sevoflurane.

Introduction 

The authors report that the antiemetics commonly used in clinical practice have adverse effects (ref 8) but these are generally mild and carry a low risk of life threatening toxicity. The primary concern is QT interval prolongation by agents, such as droperidol and ondansetron, resulting in a black box warning in the case of droperidol. Multiple experts have questioned justification for this black box warning. 

The statement: “none of these antiemetics can safely and entirely treat PONV after middle ear surgery” does not correspond with the ref 8. 

Methods

The authors should state whether the inclusion criteria identified or excluded patients previously treated with corticosteroids.

The authors should state in methods how the patient's risk factor for PONV was assessed (although the Apfel scale is well known). Since in results and in Table 1 they show the parameter "full scale" (better to make it clear in the table that it is the Apfel full scale).

What is the point of assessing the Modified Observer’s Alertness/Sedation Scale (MOAA/S) in a patient who has been administered neuromuscular blockade? This scale is typically used in patients under the effects of sedation and not under the effects of general anaesthesia and with the use of muscle relaxants. It should not replace the BIS. Data concerning the BIS during the phases of the study should be displayed.

Insufficient data have been published describing the con­centration-effect relationship of remimazolam on typical EEG pa­rameters such as the BIS during general anesthesia and during coadministration with an opioid in patients. 

In previous studies with remimazolam, which have used this scale, muscle relaxants were not used. The authors should justify the use of this scale and its validity for comparing anaesthetic depth between remimazolam and sevoflurane.

The analgesic strategy used is striking. The authors do not use any multimodal analgesia (the elimination of dexamethasone due to its antiemetic and analgesic effects is perfectly understandable), but neither paracetamol nor any NSAID is administered during surgery, but what is most striking is that ibuprofen is administered 6 hours after surgery. 

On the other hand, remifentanil was used during the operation, the analgesic effects of which disappear within a few minutes of its infusion. 

The protocol did not allow the use of opioids in the postoperative period. Does this imply that the patients did not have pain of sufficient magnitude to require another analgesic from the end of the surgery until 6 hours postoperatively? 

The authors should clarify this.

Results

In the table1  the sum of the risk factors is not clear; so if we consider that being female is a risk factor for PONV, and in the remimazolam group there were 20 females and in the sevoflurane group 23 when the total score with at least one risk factor should show 20 instead of 15 in the group remimazolam and 23 in the sevoflurane group (and not 12).

The authors should clarify these results.  

All abbreviations used should be referenced at the bottom of table 1 to make them easier to read.

Assessment of the severity of PONV by including cases of patients who did not have PONV may be redundant with respect to the analysis of the presence or absence of PONV episodes.

The overall PONV outcome is very high in all patients (43%), and in the remimazolan group it remains very high. The authors should comment on this as an incidence of almost 30% seems too high to justify the routine use of remimazolam.

Discussion 

Likewise the use of remimazolam in the present study in anesthetic induction and maintenance may resemble the PONV protective effect of propofol when used in anesthetic induction and maintenance. However, the studies conducted with midazolam evaluate its efficacy in single bolus administration (in lower dose <0.05 mg/kg) or higher dose (≥0.075 mg/kg), and at different times of anesthesia (preoperative or at the end of surgery). 

The authors should consider commenting on this.

Author Response

Response to Reviewer 1 Comments

We appreciate your kind and thorough comments and thank you for giving us the opportunity to revise and improve our manuscript based on them.

All parts of the manuscript revised based on your comments have been marked up using the ‘Track Changes’ function in MS word for your convenience and marked in red. I attached simple memo beside revised parts marked ‘Reviewer 1’.

Once again, we appreciate your comments on our manuscript and kindly ask you to review the revised manuscript.

Point 1: The authors report that the antiemetics commonly used in clinical practice have adverse effects (ref 8) but these are generally mild and carry a low risk of life threatening toxicity. The primary concern is QT interval prolongation by agents, such as droperidol and ondansetron, resulting in a black box warning in the case of droperidol. Multiple experts have questioned justification for this black box warning.

The statement: “none of these antiemetics can safely and entirely treat PONV after middle ear surgery” does not correspond with the ref 8.

 Response 1: We totally agree with you. It seems that there was a mistake in citing the reference, so we reviewed it again.

Reference [8] has been revised. Please confirm. According to the reference, despite the use of various types of antiemetics mentioned in the previous sentence, there are limitations in preventing and treating entire PONV.

  • Page 2, Line 45-47. & Page 14, Line 442-444.

Point 2: The authors should state whether the inclusion criteria identified or excluded patients previously treated with corticosteroids.

Response 2: Previous treatment with corticosteroids was included in the exclusion criteria because it could affect PONV. Missed to describe this criteria, so we revised and added about it.

  • Page 2, Line 80-81.

Point 3: The authors should state in methods how the patient's risk factor for PONV was assessed (although the Apfel scale is well known). Since in results and in Table 1 they show the parameter "full scale" (better to make it clear in the table that it is the Apfel full scale).

Response 3: We agree with your comment. It was mentioned that the ‘Apfel score’ was used to assess the PONV risk, and the method for evaluating it was additionally described. In addition, the ‘Apfel score’ was additionally marked in Table 1.

  • Page 2, Line 85-87. & Page 6, Table 1.

Points 4: What is the point of assessing the Modified Observer’s Alertness/Sedation Scale (MOAA/S) in a patient who has been administered neuromuscular blockade? This scale is typically used in patients under the effects of sedation and not under the effects of general anaesthesia and with the use of muscle relaxants. It should not replace the BIS. Data concerning the BIS during the phases of the study should be displayed.

In previous studies with remimazolam, which have used this scale, muscle relaxants were not used. The authors should justify the use of this scale and its validity for comparing anaesthetic depth between remimazolam and sevoflurane.

Response 4: We totally agree with your point about using MOAA/S. As you mentioned, published data on the correlation between remimazolam and BIS are insufficient. There was no appropriate method to evaluate the depth of anesthesia as a substitute for BIS, and the limitations of this were mentioned in the discussion.

However, it can be expected that the appropriate depth of anesthesia was maintained when considering the patient's recovery time along with MOAA/S after emergence from general anesthesia. We have added about this to the manuscript.

  • Page 13, Line 390-394.

Point 5:

The analgesic strategy used is striking. The authors do not use any multimodal analgesia (the elimination of dexamethasone due to its antiemetic and analgesic effects is perfectly understandable), but neither paracetamol nor any NSAID is administered during surgery, but what is most striking is that ibuprofen is administered 6 hours after surgery. 

On the other hand, remifentanil was used during the operation, the analgesic effects of which disappear within a few minutes of its infusion. 

The protocol did not allow the use of opioids in the postoperative period. Does this imply that the patients did not have pain of sufficient magnitude to require another analgesic from the end of the surgery until 6 hours postoperatively? 

Response 5: The postoperative analgesia protocol that was actually used was excluded from the original manuscript. As described in the manuscript, there was no statistically significant difference in postoperative NRS pain score between two groups. Also, though it was excluded from the manuscript, there was no statistically significant difference in patients' need of rescue analgesics. Therefore, we excluded the data about rescue analgesia because we thought that it would not be essential when considering the main outcome of this study.

In addition, since tympanoplasty with mastoidectomy is a surgery that causes relatively mild pain and excessive use of analgesics may affect PONV, we set our study to use minimal rescue analgesics.

However, given your comment, we thought it would be necessary to mention our rescue analgesic protocol that we actually applied. We added it by using previously collected data.

  • Page 3, Line 136-139. & Page 11, Table 4.

Point 6: In the table 1, the sum of the risk factors is not clear; so if we consider that being female is a risk factor for PONV, and in the remimazolam group there were 20 females and in the sevoflurane group 23 when the total score with at least one risk factor should show 20 instead of 15 in the group remimazolam and 23 in the sevoflurane group (and not 12).

Response 6: We do not fully understand your comment.

On the data collected in this study, PONV history and postoperative opoid use were all 0 in the county. Therefore, the total score could be derived by considering only the sex and smoking history of the two groups.

Take the Remimazolam group for example. As shown in the table below, if 5 out of 18 males are smokers and 2 out of 20 females are smokers, the total score for the Remimazolam group will be 5/15/18 (0/1/2), as shown in Table 1 of the manuscript.

Remimazolam group

Male

Female

Smoking

5 (total score 0)

2 (total score 1)

Non-smoking

13 (total score 1)

18 (total score 2)

If we misunderstood your comment, please restate it and we will gladly revise it.

  • Page 5 & 6, Table 1.

Point 7: All abbreviations used should be referenced at the bottom of table 1 to make them easier to read.

Response 7: We have revised it.

  • Page 7, Line 223.

Point 8: Assessment of the severity of PONV by including cases of patients who did not have PONV may be redundant with respect to the analysis of the presence or absence of PONV episodes.

Response 8: We understand your concern. However, we think there will be no problem in that the severity of PONV was evaluated by referring to several previous articles related to PONV.

We will attach some of the referenced articles.

  • Yuki, H.; Shiho, S.; Chiaki, M.; Soshi, N.; Atsushi, M.; Takahiro, K.; Yasuo, M.T.; Nami, K.; Katsuya, T. Remimazolam decreased the incidence of early postoperative nausea and vomiting compared to desflurane after laparoscopic gynecological surgery. Journal of Anesthesia 2022, 36, 265-69.
  • Kim, E.J.; Kim, C.H.; Yoon, J.Y.; Byeon, G.J.; Kim, H.Y.; Choi, E.J. Comparison of postoperative nausea and vomiting between remimazolam and propofol in patients undergoing oral and maxillofacial surgery: a prospective randomized controlled trial. BMC Anesthesiology 2023, 23, 132.

Point 9: The overall PONV outcome is very high in all patients (43%), and in the remimazolan group it remains very high. The authors should comment on this as an incidence of almost 30% seems too high to justify the routine use of remimazolam.

Response 9: The significance of our study is that the incidence of PONV was reduced in the remimazolam group compared to sevoflurane. However, as you commented, the incidence 28.9% is still can be considered high. Therefore, we added demonstration that further studies would be needed comparing the incidence of PONV when using remimazolam and other intravenous anesthetics in order to justify the routine use of remimazolam in consideration of PONV.

  • Page 13, Line 367-377.

Point 10: Likewise the use of remimazolam in the present study in anesthetic induction and maintenance may resemble the PONV protective effect of propofol when used in anesthetic induction and maintenance. However, the studies conducted with midazolam evaluate its efficacy in single bolus administration (in lower dose <0.05 mg/kg) or higher dose (≥0.075 mg/kg), and at different times of anesthesia (preoperative or at the end of surgery). 

The authors should consider commenting on this.

Response 10: We agree with you that our study using remimazolam differs from previous studies using midazolam in some settings. According to our best knowledge, there is no guidance of a bolus administration when remimazolam is used as a general anesthetic. There is a pharmacopeia on the use of a bolus administration when remimazolam is used for sedation during a short procedure, but there is only guidance on infusion when inducing and maintaining general anesthesia. In future studies, if possible, it would be good to find out the difference in PONV incidence according to the bolus use of remimazolam and the different timepoints of administration.

If you think further description would be needed in the manuscript after reading this reponse, please let us know and we will add and revise it.

Reviewer 2 Report

Thank you for the possibility to read this well written manuscript. The scientific setup is interesting and current. I also appreciate your accurate description of your timely, well planned and standardized anesthesia method.

Methods chapter 3: You did the best you can with the blinding of these very distinct anesthesia methods that are impossible to fully blind to the performing anesthesia team. However, as sevoflurane has its unique odor that can be smelled also in the post anesthesia care unit (PACU), do you think that the personnel in PACU and the outcome accessors are truly “fully blinded”? I suggest you edit the term in the methods and mention this in the possible limitations in the discussion.

Methods chapter 7: Could you describe how did you measure the time for hypotension periods? Was this calculated automatically during monitoring? Or observed by person?

Methods chapter 7: What was the postoperative pain care protocol? Some regional anesthesia? Some rescue analgesia? Or only ibuprofen 400mg once for all?

Methods chapter 7: Did you provide any antiemetics as prophylaxis? I understand that to see the possible beneficial effect of remimazolam this protocol is acceptable but in routine daily use it is recommendable to provide antiemetics for patients when using sevoflurane anesthesia. (If you provide prophylaxis, would you still see beneficial effects with remimazolam? This could be discussed – not obligatory)

Methods chapter 8: Did you have specific time points to evaluate the postoperative nausea and vomiting (PONV), pain etc? Did you measure the hemodynamic outcomes at T1-T5, or all the outcomes? If you present the PONV at 0-12 h after surgery, is it some specific time point? Or the average of the time points? Highest score?

Methods chapter 9: What expected SD did you use in your power calculation?

Methods chapter 9, results chapter 3, and discussion chapter 1: In methods you mention that you recognize 40% difference in the incidence of PONV as clinically significant. I think this is a very big difference expected and recognized as clinically significant. This makes your calculated sample size smaller. Thus, considering your results on PONV at 0-12h, does the result 29% remimazolam group vs 58% sevoflurane group met your own criteria (40%) of clinical significance? Is your study underpowered? You should consider this when interpreting your results and the limitations of the study in discussion. However, I still feel your study adequate as performing a large RCT is demanding, and the study still provides new information of a new anesthetic agent remimazolam.

Methods chapter 10: Is is common to present mean and SD also for nonparametric variables? You should consider median and IQR for nonparametric variables – however, is not obligatory for this paper, as the main outcome measures are binary/ categorical.

Results chapter 1 and Figure 1: Can you explain to me personally, how it is possible to evaluate 80 patients, find them all eligible for the whole study protocol (e.g. all understanding the study information, no contraindications for any of the drugs e.g. ibuprofen)? Or did you evaluate a bigger sample and describe only those, who are eligible, and you asked for informed consent? No surgery cancelations (e.g. patient sick) or any other drop-outs after recruitment or randomization? Is the data real?

Results, table 1: Your recruited 76 or 80 patients with no patients with history of motion sickness? What is the prevalence of motion sickness in your region? Did you extract the information from some medical history or ask from the patient? I understand the lack of information, but the numbers seem unreal.

Results chapter 2 and discussion chapter 4-5: You had reasonable concerns in measuring the depth of anesthesia with BIS. Further, it is impossible/ irrational to measure the depth of general anesthesia with sedation scale MOAA/S, as you used rocuronium and the patients are thus incapable to express their possible alertness. Is it possible that the differences you observed concerning hypotension are a results of differences in anesthesia depth? Is it possible that as the performing anesthetist is not blinded, it may have affected the use anesthetics/ depth of anesthesia and thus observed hypotension and use of vasoactive drugs? Please discuss with the hypotension results, and the limitations.

Results Table 4: Considering the similarity between remimazolam and midazolam and the significant amnestic effect of midazolam, how was the amnesia tested? If only grossly observed e.g. during routine clinical rounds this is understandable (not a key outcome of the study), but the limitations on this evaluation should maybe be addressed as to be evaluated by future studies.

Discussion chapter 1: You should consider revision of the text as your result on PONV does not been the clinical significance you defined in the methods section, and you did not notice any differences in vomiting

Discussion chapter 4: You should consider revision of the text, as the differences you noticed on hypotension might be associated with the problems of blinding and monitoring the depth of anesthesia

In summary: this is a well performed and described (rather small) RCT, but considering remimazolam as new agent, the study is interesting and current.

Author Response

Response to Reviewer 2 Comments

We appreciate your kind and thorough comments and thank you for giving us the opportunity to revise and improve our manuscript based on them.

All parts of the manuscript revised based on your comments have been marked up using the ‘Track Changes’ function in MS word for your convenience and marked in red. I attached simple memo beside revised parts marked ‘Reviewer 2’.

Once again, we appreciate your comments on our manuscript and kindly ask you to review the revised manuscript.

Point 1:

Methods chapter 3: You did the best you can with the blinding of these very distinct anesthesia methods that are impossible to fully blind to the performing anesthesia team. However, as sevoflurane has its unique odor that can be smelled also in the post anesthesia care unit (PACU), do you think that the personnel in PACU and the outcome accessors are truly “fully blinded”? I suggest you edit the term in the methods and mention this in the possible limitations in the discussion.

Response 1:

We understand your concern. However, it seems almost impossible for personnel in the PACU to identify the unique odor of sevoflurane in that patients are usually transferred to the PACU after confirmation of recovery to a certain level upon emergence from general anesthesia. Therefore, it is considered that full blinding is possible to the personnel in the PACU as patients delivered to the PACU would have recovered to a certain level without smell of sevoflurane.

Similar to our study, there are many articles studying the difference in PONV between using volatile anesthetics and intravenous anesthetics. To the best of our knowledge, among those other studies, we could not find any mention of limitation about poor blinding in PACU. A few references are attached below.

However, in the case of an attending anesthesiologist, as mentioned in the manuscript, blinding for the two anesthetics is not possible, so the limitations of this were additionally mentioned in discussion.

  • Page 13, Line 394-397.

  • Lee, D.W.; Lee, H.G.; Jeong, C.Y.; Jeong, S.W.; Lee, S.H. Postoperative nausea and vomiting after mastoidectomy with tympanoplasty: A comparison between TIVA with propofol-remifentanil and balanced anesthesia with sevoflurane-remifentanil. Korean J Anesthesiol 2011, 61, 399-404.

  • Bansal, T.; Singhal, S.; Kundu, K. Prospective randomized double-blind study to evaluate propofol and combination of propofol and sevoflurane as maintenance agents in reducing postoperative nausea and vomiting in female patients undergoing laparoscopic surgery. Med Gas Res 2022, 12, 137-140.

  • Kandavar, S.; Padmanabha, S. Comparison of effects propofol and sevoflurane used in maintenance of general anesthesia on post-operative nausea and vomiting – a prospective observational study. J Evolution Med Dent Sci 2021, 10, 1515-1518.

Point 2:

Methods chapter 7: Could you describe how did you measure the time for hypotension periods? Was this calculated automatically during monitoring? Or observed by person?

Response 2:

It was automatically calculated during monitoring. We have additionally mentioned it in the manuscript.

  • Page 3, Line 132.

Point 3:

Methods chapter 7: What was the postoperative pain care protocol? Some regional anesthesia? Some rescue analgesia? Or only ibuprofen 400mg once for all?

Response 3:

The postoperative analgesia protocol that was actually used was excluded from the original manuscript. As described in the manuscript, there was no statistically significant difference in postoperative NRS pain score between two groups. Also, though it was excluded from the manuscript, there was no statistically significant difference in patients' need of rescue analgesics. Therefore, we excluded the data about rescue analgesia because we thought that it would not be essential when considering the main outcome of this study.

However, given your comment, we thought it would be necessary to mention our rescue analgesic protocol that we actually applied. We added it by using previously collected data.

  • Page 3, Line 136-139. & Page 11, Table 4.

Point 4:

Methods chapter 7: Did you provide any antiemetics as prophylaxis? I understand that to see the possible beneficial effect of remimazolam this protocol is acceptable but in routine daily use it is recommendable to provide antiemetics for patients when using sevoflurane anesthesia. (If you provide prophylaxis, would you still see beneficial effects with remimazolam? This could be discussed – not obligatory)

Response 4:

I totally agree with your comment. Remimazolam is a recently licensed general anesthetic, and studies on its PONV effect are very limited. Therefore, in order to accurately confirm the possible beneficial effects of it, the use of preventive antiemetic drugs was excluded from this study.

Considering that the use of prophylactic antiemetic for PONV prevention is a recent trend, using prophylactic antiemetic in remimazolam-related PONV studies should be take in to account in further studies. We are looking forward to proceed studies in consideration of this point.

Point 5:

Methods chapter 8: Did you have specific time points to evaluate the postoperative nausea and vomiting (PONV), pain etc? Did you measure the hemodynamic outcomes at T1-T5, or all the outcomes? If you present the PONV at 0-12 h after surgery, is it some specific time point? Or the average of the time points? Highest score?

Response 5:

The incidence rate of PONV was based on one or more nausea or vomiting episodes as mentioned in the manuscript. In the case of the severity of PONV, the highest nausea score in the corresponding period was used as the criterion.

In the case of hemodynamic data, it was actually measured every 5 minutes, but the assessment was conducted only at T1-T5.

  • Page 3, Line 145-152.

Point 6:

Methods chapter 9: What expected SD did you use in your power calculation?

Response 6:

When calculating the sample size based on two proportions, we need proportion for each group. Standard deviation could be estimated by formula using binomial distribution, but it is not the element required in power calculation.

Point 7:

Methods chapter 9, results chapter 3, and discussion chapter 1: In methods you mention that you recognize 40% difference in the incidence of PONV as clinically significant. I think this is a very big difference expected and recognized as clinically significant. This makes your calculated sample size smaller. Thus, considering your results on PONV at 0-12h, does the result 29% remimazolam group vs 58% sevoflurane group met your own criteria (40%) of clinical significance? Is your study underpowered? You should consider this when interpreting your results and the limitations of the study in discussion. However, I still feel your study adequate as performing a large RCT is demanding, and the study still provides new information of a new anesthetic agent remimazolam.

Response 7:

I fully agree with your point. As mentioned in the manuscript, the sample size was calculated based on the fact that the use of midazolam, a bezodiazepine like remimazolam, can reduce PONV by 40%. When considering data results of our study, we believe that our criterion was satisfied.

However, as you commented, the sample size of our study is considered to be too small, and this was mentioned in discussion as a limitation of the study. Based on our study, we expect to proceed with larger RCTs later.

  • Page 13, Line 378-380.

Point 8:

Methods chapter 10: Is is common to present mean and SD also for nonparametric variables? You should consider median and IQR for nonparametric variables – however, is not obligatory for this paper, as the main outcome measures are binary/ categorical.

Response 8:

We have added median and IQR for nonparametric variables. We would like you to check the additions in the tables.

  • Page 5/6/7, Table 1 & Page 11, Table 4.

Point 9:

Results chapter 1 and Figure 1: Can you explain to me personally, how it is possible to evaluate 80 patients, find them all eligible for the whole study protocol (e.g. all understanding the study information, no contraindications for any of the drugs e.g. ibuprofen)? Or did you evaluate a bigger sample and describe only those, who are eligible, and you asked for informed consent? No surgery cancelations (e.g. patient sick) or any other drop-outs after recruitment or randomization? Is the data real?

Response 9:

We fully understand your concern. As mentioned in the methods, we targeted a total of 80 patients, of which 4 (2 morbid obesity, 2 lack of consent) were excluded due to criteria, resulting in a study of 76 patients in total.

All 76 patients finally included in the study provided informed consent. As mentioned in the inclusion and exclusion criteria, only elective surgery was included, so it is thought that almost all patients were eligible. In addition, due to the characteristics of middle ear surgery, this result seems to have been derived because unusual circumstances such as cancellation of surgery (e.g. patient sick) are rare.

We understand you may have doubts about the data. However, we do not consider that this result is impossible to be obtained due to the reasons mentioned above. We are sure that all the progress of the study was carried out as described in the methods.

Point 10:

Results, table 1: Your recruited 76 or 80 patients with no patients with history of motion sickness? What is the prevalence of motion sickness in your region? Did you extract the information from some medical history or ask from the patient? I understand the lack of information, but the numbers seem unreal.

Response 10:

We agree to some extent with your point. When referring to several studies, including the reference to be attached below, it is known that motion sickness occurs in approximately one-third of the population. In the case of patients enrolled in our study, data came out that there was no patient with motion sickness history in both groups. This is a result that is far from the known prevalence of motion sickness, as you are concerned about.

We accept that there may be data errors due to limited information, and we will pay more attention to data collection in future studies. Although the number of data may seem unrealistic, we promise that we have obtained data based on patient interviews along with checking patients' medical records.

During routine clinical rounding, we interviewed patients. The contents of the questionnaire included the following contents.

(Motion sickness is a common condition characterized by a feeling of unwellness brought on by certain kinds of movement. The usual symptoms include dizziness, pallor, and sweating, followed by nausea and vomiting. Affected individuals may also experience hyperventilation, headache, restlessness, and drowsiness.)

  • Sherman, C. R. Motion sickness: review of causes and preventive strategies. Journal of Travel Medicine 2006, 9, 251–6.

Point 11:

Results chapter 2 and discussion chapter 4-5: You had reasonable concerns in measuring the depth of anesthesia with BIS. Further, it is impossible/ irrational to measure the depth of general anesthesia with sedation scale MOAA/S, as you used rocuronium and the patients are thus incapable to express their possible alertness. Is it possible that the differences you observed concerning hypotension are a results of differences in anesthesia depth? Is it possible that as the performing anesthetist is not blinded, it may have affected the use anesthetics/ depth of anesthesia and thus observed hypotension and use of vasoactive drugs? Please discuss with the hypotension results, and the limitations.

Response 11:

We totally agree with your point about using MOAA/S. As you mentioned, published data on the correlation between remimazolam and BIS are insufficient. There was no appropriate method to evaluate the depth of anesthesia as a substitute for BIS, and the limitations of this were mentioned in the discussion.

However, it can be expected that the appropriate depth of anesthesia was maintained when considering the patient's recovery time along with MOAA/S after emergence of general anesthesia. We have added about this to the manuscript.

In addition, we added demonstration that since the attending anesthesiologist was not blinded, it might have affected the depth of anesthesia and hemodynamic data.

  • Page 13, Line 380-397.

Point 12:

Results Table 4: Considering the similarity between remimazolam and midazolam and the significant amnestic effect of midazolam, how was the amnesia tested? If only grossly observed e.g. during routine clinical rounds this is understandable (not a key outcome of the study), but the limitations on this evaluation should maybe be addressed as to be evaluated by future studies.

Response 12:

We agree with your comments. As you said, data were collected through patient assessments and patient interviews during routine clinical rounds after surgery. Amnesia as well as all other postoperative complications listed in the manuscript were identified as well. As you commented, since this was not the key outcome of our study, we thought there was no need to address the limitations and methods for identifying postoperative complications, including amnesia.

Nevertheless, if you think that additional descriptions of amnesia should be included in the manuscript, we will gladly add them.

Point 13:

Discussion chapter 1: You should consider revision of the text as your result on PONV does not been the clinical significance you defined in the methods section, and you did not notice any differences in vomiting

Response 13:

Unfortunately we didn't fully understand your comment. The primary and secondary outcomes defined in the methods and the outcomes described in the results appear to be the same. As mentioned in methods, we anticipated that a 40% lower incidence rate of PONV when using remimazolam compared to using sevoflurane would be clinically significant, and the results of our study actually met this criteria. In the case of the incidence rate of vomiting 0-48 h after surgery, there was no statistically significant difference as described in results, but since vomiting is also thought to be included in PONV, the discussion only mentioned the incidence rate and the severity of PONV at each timepoint.

If our answers fall short, please let us know. We will make further revisions if necessary.

Point 14:

Discussion chapter 4: You should consider revision of the text, as the differences you noticed on hypotension might be associated with the problems of blinding and monitoring the depth of anesthesia

Response 14:

Some limitations were additionally described in consideration of blinding, anesthetic depth, and hemodynamic stability.

  • Page 13, Line 380-397.